# Alpha-Gal and Cross-Reactive Carbohydrate Determinants in the N-Glycans of Salivary Glands in the Lone Star Tick, *Amblyomma americanum*

**DOI:** 10.3390/vaccines8010018

**Published:** 2020-01-09

**Authors:** Yoonseong Park, Donghun Kim, Gunavanthi D. Boorgula, Kristof De Schutter, Guy Smagghe, Ladislav Šimo, Stephanie A. Archer-Hartmann, Parastoo Azadi

**Affiliations:** 1Department of Entomology, Kansas State University, Manhattan, KS 66506, USA; dklome2018@knu.ac.kr (D.K.); boorgulagd@ksu.edu (G.D.B.); 2Department of Applied Biology, Kyungpook National University, Sangju 37224, Gyeongbuk, Korea; 3Department of Plants and Crops, Faculty of Bioscience Engineering, Ghent University, Coupure Links 653, 9000 Ghent, Belgium; kristof.deschutter@ugent.be (K.D.S.); Guy.Smagghe@UGent.be (G.S.); 4UMR BIPAR, INRAE, Ecole Nationale Vétérinaire d’Alfort, ANSES, Université Paris-Est, 94700 Maisons-Alfort, France; ladislav.simo@vet-alfort.fr; 5Complex Carbohydrate Center, University of Georgia, Athens, GA 30602, USA; sarcher@ccrc.uga.edu (S.A.A.-H.); azadi@ccrc.uga.edu (P.A.)

**Keywords:** alpha-gal, red meat allergy, glygosylation, xylosylation, molecular mimicry, glycan

## Abstract

Ticks are important ectoparasites and vectors of numerous human and animal pathogens. Ticks secrete saliva that contains various bioactive materials to evade the host defense system, and often facilitates the pathogen transmission. In addition, the Lone star tick saliva is thought to be the sensitizer in red meat allergy that is characterized by an allergic reaction to glycan moieties carrying terminal galactose-alpha-1,3-galactose (aGal). To assess N-glycome of *Amblyomma americanum*, we examined the N-glycan structures in male and female salivary glands at three different feeding stages and in carcasses of partially fed lone star ticks. We also surveyed the genes involved in the N-glycosylation in the tick species. The aGal epitopes and cross-reactive carbohydrate determinants (CCD) increases over time after the onset of blood feeding in both male and female *A. americanum*. These CCDs include xylosylation of the core mannose, 1,3-mono and 1,3- and 1,6-difucosylations of the basal GlcNac and mono- or diantennary aGal. Combinations of both xylosylation and aGal and fucosylation and aGal were also found on the N-glycan structures. While the enzymes required for the early steps of the N-glycosylation pathway are quite conserved, the enzymes involved in the later stages of N-glycan maturation in the Golgi apparatus are highly diverged from those of insects. Most of all, we propose that the aGal serves as a molecular mimicry of bioactive proteins during tick feedings on mammalian hosts, while it contributes as a sensitizer of allergy in atypical host human.

## 1. Introduction

Protein glycosylation, or the attachment of (oligo)saccharides to the amino acid residues Asn (*N*-glycosylation) and Ser or Thr (*O*-glycosylation), is the most common and important post-translational modification. Protein glycosylation has important roles in protein maturation and interactions with other molecules and influences properties and activities of the protein [1]. Protein glycosylation is essential in all eukaryotic cells and is involved in multiple crucial processes in the cell. This is reflected in the conservation of glycosylation sites and glycan structures [2,3] and essential roles of the enzymes involved in glycan biosynthesis and maturation [4,5]. While the metabolic pathways for the early steps of N-glycosylation are well conserved throughout different taxa of eukaryotes, divergence is observed at the subsequent steps generating interspecies and intraspecies specific N-glycan structures [6]. This diversity of glycans in different organisms involves evolutionary selection processes with the selection pressure on the presence/absence of functional metabolic genes, enzyme activities, and their substrate specificities. The general evolutionary pattern of N-glycan likely involved drift in neutral or near-neutral evolution based on the high variations in closely related taxa, whereas the parts of N-glycan variations were consequences of selection pressure in the arms races between pathogen and immune systems [7].

An interesting aspect of glycan function is its role in pathogen (or commensal bacteria) recognition and vector-host interactions. In the immune system, which consists of a complex network of cells and molecules that interact with each other to initiate the host defense system, many interactions involve specific carbohydrate structures and proteins that specifically recognize and bind them [8]. So are pathogens recognized through highly conserved structures, pathogen associated molecular patterns (PAMPs) [9]. Many of these PAMPs are carbohydrate structures present in the cell wall or glycan structures present on the cell surface, which are bound by the pattern recognition receptors (PRRs) of the host immune system [9]. On the other hand, the binding of pathogens to host cells in infection processes often depends on the glycan receptors on the host cell surface [10]. In the evolutionary arms race, some pathogens adopted the strategy of “molecular mimicry” and decorate their surface with host glycan structures to evade the host immune system. For example, the incorporation of sialic acid (Sia) in the gram negative bacterium *Pseudomonas aeruginosa* allows binding with the Sia-binding immunoglobulin-like lectins (siglecs) on the surface of neutrophils [11]. This interaction causes the neutrophil to recognize the bacteria as ‘self’ which dampens the neutrophil reactivity [12]. Molecular mimicry of host glycans is also known in *Trypanosoma cruzi*, which utilizes host sialic acid transferred to the terminus of mucin-like *O*-glycan on the surface of the parasite [13,14]. Further, molecular mimicry has been identified as one of the mechanisms by which infections can initiate or exacerbate autoimmune diseases [15].

Ticks are important hematophagous arthropods and vectors of numerous human and animal pathogens. These external parasites mainly belong to two families, the Ixodidae (hard ticks, ~700 species) and the Argasidae (soft ticks, ~200 species) [16]. During blood feeding, ticks secrete saliva that contains various bioactive materials to evade the host defense system and they can transmit various pathogens causing diseases, e.g., Lyme disease and tick-borne encephalitis [17]. Recently, glycomics of ticks have attracted attention due to certain species of ticks, including the lone star tick *Amblyomma americanum* (Ixodidae) in the U.S, were found to be causing red meat allergy (RMA). RMA in humans, which is diagnosed by a 3–5 h delayed urticaria and anaphylaxis to dietary red meat, is caused by an unusual occurrence of enhanced IgE antibody production against glycan moieties carrying terminal galactose-alpha-1,3-galactose (aGal). This is a common glycan in mammalian tissue with the exceptions of old world monkeys and humans [18,19,20,21,22].

In this study, we investigated the *N*-glycans in male and female lone star ticks *A. americanum* over a duration of blood feeding. Based on the results showing high levels of aGal in the salivary glands (SGs) during blood feeding, we propose that ticks use aGal as a molecular mimicry to evade the immune system of nonprimate hosts, while the aGal may function as an allergy sensitizer in the atypical host human.

## 2. Materials and Methods

### 2.1. N-Glycan Profiling

Adult ticks were purchased from Oklahoma tick rearing center. Partially engorged female ticks (4–5 days blood fed) were prepared using a modified artificial feeding system [23,24]. Dissected tissues from ~50 individual ticks were made over a duration of a month in multiple feeding chambers, and were stored at −80 °C until the analyses.

The overall workflow for glycan analysis included the following steps: homogenization, tryptic digestion, N-glycan release, permethylation, and nanospray ionization–mass spectrometry (NSI-MS) including tendem mass spectrometry (MS/MS) fragmentation. All reagents were purchased from Sigma Aldrich unless otherwise mentioned. Mass spectrometric data acquisition was performed on a Thermo Scientific LTQ Orbitrap Fusion Tribrid mass spectrometer. Data analysis was performed by using Data Explorer V4.5, Xcalibur 3.0, and GlycoWorkbench 1.1.

### 2.2. Release of N-Linked Glycans

For the samples, ~50 pairs of dissected salivary glands or 20 carcasses, were homogenized in PBS for 30 s (Branson Sonifier), reduced for 45 min at 45 °C in a solution of 5 mM DTT (Dithiothreitol), and alkylated at room temperature in the dark in a solution of 15 mM iodoacetatmide. A subsequent aliquot of DTT (5 mM) was added to quench the reaction. The reduced and alkylated samples were then digested with trypsin (porcine, Sigma) in 50 mM pH 8 Tris-HCl buffer overnight. After protease digestion, the sample was passed through a C18 Sep-Pak cartridge (Supelco), washed with 5% *v*/*v* acetic acid in mΩ water (Barnstead), and the glycopeptides were eluted with a blend of isopropanol (40%, 80%, and 100%) in low 5% *v*/*v* acetic acid before being dried by SpeedVac [25].

The dried glycopeptide eluate was digested with a combination of PNGase A (Calbiochem) and PNGase F (New England Biolabs) overnight at 37 °C to release the N-linked glycans. The digest was then passed through a C18 Sep-Pak cartridge (Supelco) with 5% *v*/*v* acetic acid to recover the N-glycans. The pass-through fraction containing released glycans was then dried via lyophilization.

### 2.3. Per-O-Methylation of N-Linked Glycans

The glycans were permethylated for analysis by ESI-MS (Electrospray ionization mass spectrometry). Briefly, the sample was dissolved in dimethyl sulfoxide (DMSO) and incubated with methyl iodide in a DMSO/NaOH mixture. The reaction was quenched with water, and permethylated O-glycans were extracted with methylene chloride.

### 2.4. Profiling and Fragmentation Analysis by Nanospray Ionization-Mass Spectrometry (NSI-FTMS/MS)

The sample was then dissolved in methanol, and an aliquot was taken for NSI-MS. ESI-MS and MS/MS analysis of permethylated glycans was performed on an Orbitrap Fusion mass spectrometer through an NSI probe [25]. The MSn spectra collision-induced fragmentation (CID) and higher-energy collisional dissociation (HCD) of the glycans were acquired at high resolution (selected CID spectra are shown since HCD produced similar spectra). Assignment of glycan structures was performed manually, and with assistance by GlycoWorkbench software.

Full MS spectra as well as an automated “TopN” MS/MS program of the top 300 peaks were collected and fragmented with CID. These data were used to search for a Hex-Hex-HexNAc signature, both with a diagnostic fragment and expected neutral losses.

When glycoforms matching a Hex-Hex-HexNAc signature were found in some of the samples, the rest of the samples were rerun and manually fragmented at masses corresponding to each of the possible N-linked glycoforms. This was performed to confirm the presence or absence of these glycoforms even if they were in trace amounts and not detected in the full MS spectra.

### 2.5. Bioinformatics

Using the NGRG protein sequences from *D. melanogaster*, *T. castaneum* and humans as queries in NCBI BLAST searches, orthologous sequences were identified in the tick proteomes or genomes (taxid:6933). Identified sequences were used for back-blasts to the databases of *D. melanogaster*, *T. castaneum* and human sequences and analyzed for domain structure through Motif search (https://www.genome.jp/tools/motif/) to verify their orthology.

The set of putative NGRG sequences obtained in the blast searches were used for sequence alignment and phylogenetic analyses with the known *D. melanogaster*, *T. castaneum* and human sequences. The protein sequences were aligned with MUSCLE (Multiple Sequence Comparison by Log-Expectation) using default settings in MEGA7 (Molecular Evolutionary Genetics Analysis) [26]. Neighbor-joining trees [27] were build and a phylogenetic tree is drawn with the relative scale of branch lengths computed using the Poisson correction method. Five hundred bootstrapping values are shown next to the branches [28].

## 3. Results

### 3.1. Spatial and Temporal Changes in the N-Glycosylation Patterns in the Salivary Glands of Male and Female A. americanum

To explore the effect of feeding on the N-glycan profile of the lone star tick, male and female ticks were allowed to feed in an artificial membrane feeding chamber [23,24]. Three different time points relative to the blood feeding of the ticks were chosen for the analyses of the N-glycan profile in the SGs: before the onset of feeding, two days after, and five days after feeding attachments (0dAF, 2dAF, and 5dAF). Male ticks of this species are known to feed multiple times for short durations [29]. Therefore, the 2dAF samples represent the timepoint after completion of the first feeding, and the 5dAF samples represent the timepoint in or after the second attachment in the male feeding. For females, the initial feeding is usually continued until complete engorgement. The full engorgement of *A. americanum* females took approximately 10 days of blood feeding. Therefore, the samples 5dAF are in the middle of the female feeding period and ticks are significantly bigger in their sizes.

N-glycan profiles in the lone star tick were determined for the SGs at the three timepoints described above and for the tick carcasses (CCs) at 5dAF. N-glycans were released from polypeptides by PNGase A and PNGase F and subsequently identified by Matrix-Assisted Laser Desorption/Ionization-Mass Spectrometry (MALDI-MS) and ESI-MS. Clustering analysis of the different samples based on the N-glycosylation pattern showed two main clusters based on the tissue of the samples (Figure 1). In the analyses, same molecular weights containing mix of two different glycan were corrected by normalization factor ½ in the percent quantity (i.e., 2244.1 and 2489.3). The samples from the male and female CCs grouped together in a cluster separated from the SG samples (Figure 1, full list is in the Appendix A). Within the cluster of the SG samples, the samples of male and female ticks at 5dAF and 2dAF grouped based on the timepoint (Figure 1). The N-glycan profile of the CC samples was dominated by paucimannose structures. While female CCs mainly contained non-fucosylated paucimannoses (77% of the total N-glycan profile), the majority of the paucimannose N-glycans in male CCs were monofucosylated with a high portion of fucosylation at the basal N-acetylglucosamine (GlcNac) (61% of the total N-glycan profile) and about 17% of the total N-glycan profile was di-fucosylated (Appendix A). Except for a minor fraction (2% of the total N-glycan profile) in males, the CC samples lacked high mannose and complex glycans, while the SG samples were generally rich in high mannoses and complex glycans. The Shannon diversity H and the richness D progressively increased in the 2dAF and 5dAF SG samples in both males and females due to increases in complex N-glycan structures (Figure 1 and Figure 2). This increase in the diversity of the N-glycans in the SG samples is due to by the high proportion of complex N-glycans (i.e., 2244.1, 2285.1, and 2489.3 in Figure 1, Appendix A).

In addition, the presence of N-glycans containing galactose alpha-1,3-galactose beta-1,4-N-acetylglucosamine structures, which is the most common aGal and is known as the causal factor of RMA, increased over time in the SG samples during the different feeding stages (Figure 2B). Interestingly, xylosylation at the core mannose was also commonly found in the SG samples of unfed females (8% of the total N-glycan profile) and 5dAF females (5% % of the total N-glycan profile) and males (16% of the total N-glycan profile) (Figure 2B, Appendix A). This modification of core mannoses with xylose was mainly found in plants and some parasitic organisms [30] and is considered an important cross-reactive carbohydrate determinant (CCD) in allergens [31]. Similar to insect species, paucimannose with core −1, −3 and 1, 6-fucose were commonly found (Figure 1). In depth analysis of the monofucosylated paucimannose N-glycans by their MS decay patterns, these N-glycan structures were mainly with alpha-1,3-fucose (*m*/*z* 1011.5, 1133.6, 1141.6, 1235.6, and 1345.7, i.e., Appendix A).

A proportion of N-glycan structures identified in the SG samples contained both a core xylose and an aGal modification or both a core mono-fucose and aGal modification (Figure 3). However, N-glycans carrying both a core xylose and a fucose modification were not detected. Significant portions of aGal glycans were found to be di-aGal and contained the core fucose (Figure 3). Other trace glycans (<0.5% of the total ions) containing N-acetylneuraminic acid and N-glycolylneuraminic acid at the terminal galactose were observed in the SG samples of 2dAF females and males, respectively (Figure 4).

### 3.2. Genes Encoding the Enzymes for N-Glycan Processing

Protein N-glycosylation is enabled via a complex and integrated sequence of enzymatic reactions shown in the diagram in Figure 5. The putative N-glycosylation related genes (NGRGs) in the genome of ticks, encoding for the enzymatic activities of the N-glycosylation pathway, were identified through BLAST searches using the known NGRG sequences from *Drosophila melanogaster*, *Tribolium castaneum* and *Homo sapiens*. A phylogenetic analysis was performed on the identified NGRG sequences of eight tick species (*Amblyomma americanum*, *A. cajennense*, *A. meculatum*, *A. sculptum*, *A. triste*, *A. variegatum*, *Ixodes ricinus*, and *I. scapularis*), *Drosophila melanogaster*, *Tribolium castaneum* and *Homo sapiens* to analyze their evolutionary relationship. The early steps of N-glycan biosynthesis start in the endoplasmic reticulum (ER) by the transfer of the fourteen sugars N-glycan precursor from its dolichol phosphate carrier to an Asn residue of the growing peptide by the oligosaccharide transferase (OST) complex. This step is highly conserved and an ancestral character. Ticks have a highly conserved one-to-one orthology with the OST subunits STT3A and STT3B and the dolichyl-diphospho-oligosaccharide-protein glycosyltransferase subunit DAD1 (Figure 5 and Figure 6 (1)). The next step in the N-glycosylation pathway is the trimming of the terminal glucose, an essential step required for the correct folding and the quality control of the glycoproteins. This crucial role of the glucose mannosyl-oligosaccharide glucosidases GCS1 and 2, is translated in the conservation of the genes encoding these activities with insect genes (Figure 6 (2)).

The enzymes involved in the next steps of the pathway, trimming of alpha-1,2-mannoses (Step 3) and addition of a GlcNAc residue to the core alpha-1,3-mannose (Step 4) (Figure 5), appear to have a moderate degree of complexity. At least five different genes were grouped for the Class-I *α*-mannosidases (Man1a and b) that trim Man_9_GlcNac_2_ to Man_5_GlcNac_2_. Four tick genes were orthologous to those of insects, while one group was specific only in the ticks (Figure 6 (3)). The N-acetylglucosaminyltransferase, which adds a GlcNac on trimmed mannose, could be subcategorized into three subgroups: Mgat1, alpha-1,3-mannosyl-glycoprotein 2-beta-N-acetylglucosaminyltransferase; Mgat2, alpha-1,6-mannosyl-glycoprotein 2-beta-N-acetylglucosaminyltransferase; and Mgat4, alpha-1,3-mannosyl-glycoprotein 4-beta-N-acetylglucosaminyltransferase. The *I. scapularis* genome contained two predicted genes for each Mgat1 and Mgat2 and one orthologous gene for Mgat4 (Figure 6 (4)).

The addition of the first GlcNAc residue is required for the further trimming of mannose in the Golgi by Class II *α*-mannosidases (Steps 5 and 7 in Figure 5). Ticks have two orthologues of insect Class II *α*-mannosidases and at least 4 additional genes in this group in the case of *I. scapularis* (Figure 6 (5) and Figure 7). Other ticks also have additional genes in this group, but because the genomes and transcriptomes of other tick species are not complete, a clear phylogenetic analysis of this group of genes could not be provided. A characteristic to the N-glycome of invertebrates is the presence of paucimannoses [32], formation of these N-glycan structures consisting of (un)modified tri- or bimannosylchitobiosyl cores requires the activity of hexosaminidases which remove the non-reducing N-acetylglucosamine (GlcNAc) (Step 6 in Figure 5). The phylogeny of hexosaminidase indicated rapid evolution in this group of genes. Five different genes in this group were found in *I. scapularis*, but as the paralogues of insect hexosaminidases (Figure 6 (6)).

Mono- or di-fucosylated N-glycans are commonly found in invertebrate N-glycomes (Zhu et al., 2019) (Steps 8, 9, and 11 in Figure 5). Large numbers of fucosyltransferase homologues were identified in the tick species (Figure 6 and Figure 7. Steps 8, 9 and 11). Invertebrates can modify their N-glycans with alpha-1,3- and alpha-1,6-fucose [33]. One set of genes in *I. scapularis* was found to be orthologous to human FUT8, or alpha-1,6-fucosyltransferase. Additionally, another set of fucosyltransferase homologues, which are likely grouped with alpha-1,3-fucosyltransferase, were identified in the tick species. Large numbers of galactosyltransferases were identified, and their biochemical activities were recently studied [34]. Based on this study, the 57 genes are homologous to alpha-1,4-, beta-1,3- and beta-1,4-galactosyltransferases without alpha-1,3-galactosyltransferase orthologues in ticks. However, this study claimed that the genes in orthologous groups of beta-1,4- and alpha-1,4- galactosyltransferases showed alpha-1,3-galactosyltransferase activity for production of the terminal aGal productions. Two groups of sialyltransferases for each 2,3-sialylation and 2,6-sialylation were found in ticks with reliable orthologous relationships with insects and human sialyltransferases (Figure 7 (11)).

Overall, a number of enzymes showed high to moderate levels of conservation by having orthologous in insects and human enzymes (i.e., Steps 1, 2, 3, 4, and 11 in Figure 6 and Figure 7). However, tick species appear to have recent gene expansions for Class II mannosidases, hexosaminidases, alpha-1,6-fucosyltransferases, alpha-1,3-fucosyltransferases, and galactosyltransferases (Steps 5, 6, 8, and 9 in Figure 6 and Figure 7) with rapid evolution. This conclusion is yet based on the gene repertoire of *I. scapularis*, providing the most comprehensive gene sets in the genome project, while other tick species also show general patterns supporting the conclusion, but with limited numbers in the search results.

## 4. Discussion

### 4.1. Dynamic Control of the N-Glycosylation Patterns in Tick SGs

Analysis of the N-glycan profile of the carcass samples (CC) of the ticks revealed some similarity to insect glycomes. The CC N-glycome of the ticks was dominated by paucimannose N-glycans of which a significant proportion is core GlcNAc fucosylated. While the majority of the N-glycans in female CCs were non-fucosylated, in male ticks, the mono-fucosylated paucimannosed is the largest group and 17% of the total N-glycan pool is even difucosylated. This modification of the core GlcNAc with *α*1,3- and *α*1,6-fucoses is commonly found in insect N-glycans (reviewed in [2,35]), including phospholipase A_2_ and hyaluronidase of bee venom [36], and glycosylations in *D. melanogaster* [37,38,39]. While in insects oligomannose and (core fucosylated) paucimannosidic N-glycans are the most common structures [37,38,40], oligomannose structures were not identified in the CC samples of the ticks. In the SGs from unfed ticks, oligomannoses are abundantly present. An interesting observation is the difference in the fucosylation levels between male and female ticks, which is associated with the increase in mono- and di-fucosylated paucimannose N-glycans in males ticks compared to female ticks. Similar pattern of difference was also observed between the N-glycome of female and male samples of the pest insect *Nilaparvata lugens* [3]. Analysis of the N-glycome of the unfed SG showed a decrease in abundance of (core fucosylated) paucimannoses, in contrast, there is a high abundance in oligomannose and complex-type N-glycans, leading to increased diversity and richness.

This increased diversity and richness of the SG N-glycans is mainly due to different numbers of terminal galactose on the core complex glycans. In addition, xylosylation in the core mannose, different branch numbers (two to four), and a small portion of sialylation at the terminal galactose (N-acetylneuraminic acid and N-glycolylneuraminic acid) were the source of the increased diversity in the SGs of later stages of tick feeding. Many complex glycans found in the tick SGs are glycans commonly found in mammals.

Tick SGs are dynamically controlled for the production of saliva [17], which has various bioactivities including modulating and eventually evading the host immune system. In addition, a large portion of the tick salivary secretion at the later stages of tick feeding was found to contain host proteins [41,42]. Therefore, the increased diversity of N-glycan in tick SGs over the duration of feeding is likely consistent with the changes in the repertoire of the SG proteins, potentially including the host proteins with N-glycan at the late feeding stage. The N-glycan patterns of tick SGs indicates that the saliva secreted into the host may use aGal as a molecular mimicry decorating the proteins to evade the host immune system as the cases of sialic acids mentioned in the introduction. Decoration of the tick salivary proteins with the host-specific glycan aGal would be an efficient way of evading the host immune system in the case of parasitization of non-primate mammals (i.e., goat, sheep, horse, mouse, pig, and deer) that use aGal in the intrinsic system [43,44], although it may unexpectedly function as an allergy sensitizer in humans lacking aGal.

### 4.2. Cross-Reactive Carbohydrate Determinants (CCDs) and aGal in Tick N-glycans

Glycomics of ticks provide interesting datasets on the potential effects of ticks on the host immune system. Identification of the N-glycan structures in the tick SGs revealed several CCDs associated with severe allergic reactions [30,31,45,46]. One of these is the fucose epitopes characteristic to insects. Fucosylation of the core GlcNAc by alpha-1,3-fucose or difucosylation with alpha-1,3- and alpha-1,6-fucoses, known as the CCDs of hymenopteran venom (reviewed in [2,35]), were abundantly found in the tick SGs and CC samples. Interestingly, a study found high incidence of allergy to insect venom in aGal allergy patients [47]. Tick saliva with both allergens may be responsible for the occurrence of allergy to both insect venome and aGal allergy.

Xylose residues attached at the core mannose, a CCD found in peanut [31], was also commonly discovered in the tick SGs. In addition, a trace amount of N-glycan structures carrying the terminal N-glycolylneuraminic acid, another CCD [31], was also detected in the SGs of partially fed ticks (Figure 4). Several different CCDs are shown to interact with specific IgE (sIgE), which can mediate a hypersensitive immune response leading to allergies to certain plants and insect venoms [48]. However, clinical importance of the CCD as the causal factors of allergy is yet unclear, while correlation between sIgE activities against the specific CCD and the clinical symptoms has been shown [31,49].

### 4.3. aGal in the Tick SG as the Sensitizer of Red Meat Allergy

Red meat allergy (RMA) is an allergic reaction to aGal caused by a hypersensitive IgE response upon recognition of the glycan epitope. Although aGal has been well documented as an elicitor of RMA, the diagnostic test also suffers from sIgE-based assays for false-positive results [50,51,52,53,54,55,56,57]. The aGal allergy is also highly relevant to hyperreactions to xenoplantation [58] and to anti-cancer therapy using the monoclonal antibody cetuximab [59]. In all mammals except old world monkeys and humans, aGal is produced by an alpha-1,3-galactosyltransferase, which is mutated in humans [20,60]. A recent study proposed that RMA is elicited by glycolipids containing aGal but not by glycoproteins due to the difference in transport across the gut cell layer [61]. However, the nature of RMA sensitizers, likely originating from tick salivary secretion, needs to be further investigated.

Tick saliva is strongly supported as the factor leading to sensitization to aGal [21,53,62]. Based on our study and in previous studies [53], certain tick species contain aGal in the salivary secretion, which can be considered to be the sensitization factors for RMA. Our study found that upon feeding, the amount of aGal is progressively increased up to 16% and 8% in the SGs of *A. americanum* females and males, respectively. Some variants of aGal were di-aGal carrying aGal at both termini of biantennary glycan. In addition, N-glycan structures were identified containing more than one CCD together in a single glycan, structures with a terminal aGal and xylosylation of the core mannose or a terminal aGal and core fucosylations were commonly found in the SGs. Whether these N-glycans containing multiple CCDs can be recognized by separate sIgEs, facilitating cross-linking and aggregation of Fc-epsilon-RI (FcεRI) as an allergic reaction, is yet unknown.

Despite the presence of aGal epitopes in the N-glycome of *I. scapularis* saliva [53], it is unlikely that bites from this tick are causing RMA. *I. scapularis*, the vector for *Borrelia burgdorferi*, which causes Lyme disease, is found in a region including the northeastern United States, while RMA is mainly found in southeastern US, where *A. americanum* is the major tick. This discrepancy raises the question whether aGal in the salivary N-glycome of the tick alone is the RMA sensitization factor. In addition, as mentioned above, the sIgE-based diagnostic assay found that the assay results included a substantial portion of false-positive results and difficulties in prediction of the allergic clinical symptoms [51,56,57].

The source of aGal in tick SGs has been recently studied. A number of galactosyltransferase homologs in ticks are involved in aGal production in tick SGs [34,54]. An alternative source of aGal could be the N-glycans on the host proteins. Tick saliva contains large numbers of host proteins [41,42], including the host immunoglobulins [63,64]. Therefore, we included male *A. americanum* in this study, which engages in multiple intrastadial feedings (our observation and [29]), unlike the females [65], which exhibit much greater fidelity for a single feeding event for a long duration on a single host. Similar to the female *A. americanum*, the male was also found to carry aGal in the SGs in this study. Males having multiple intrastadial feeding may have a higher opportunity to switch feeding hosts from nonhuman mammals to humans, while the aGal transmission or the aGal production in the SG is preactivated.

Many important pathogens including arthropod borne pathogens have been found to carry aGal including arthropod borne pathogenes, such as *Tripanosoma* and *Leshmania* [66,67], *Borrelia* [68], and Dengue virus derived from mosquito cell [69]. More importantly, the fact that aGal functions as the protective antigen against Plasmodium, Tripanosoma, and Leshmania [70,71,72] led a basis to propose a single antigen pan-vaccine [68,73]. It would be interesting if those pathogens and the vectors commonly utilize the aGal as the molecular mimicry and the processes can be manipulated as a mean of the pathogen prevention.

## 5. Conclusions

The N-glycosylation patterns in the tick SGs change during tick feeding. The aGal epitopes, the presumed sensitizer of red meat allergy, increases over time after the onset of blood feeding. The aGal may serve as a molecular mimicry of bioactive proteins during tick feedings on mammalian hosts, while it functions as an allergen in non-natural host human.

N-Glycans of tick SGs proteins contain multiple CCDs: xylosylation of the core mannose, 1,3-mono and 1,3- and 1,6-difucosylations of the basal GlcNac and mono- or dianteneary aGal. Combinations of both xylosylation and aGal and fucosylation and aGal were often found in an N-glycan.

While the enzymes required for the early steps of the N-glycosylation pathway are conserved, the enzymes involved in the later stages of N-glycan maturation in the Golgi apparatus are highly diverged from those of insects.

## Figures and Tables

**Figure 1 vaccines-08-00018-f001:**
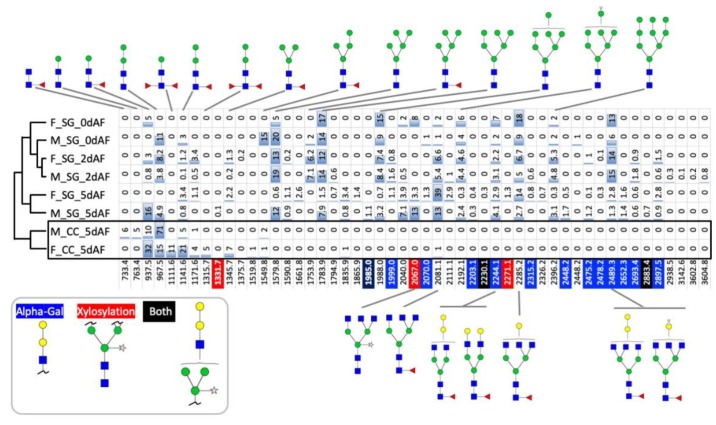
Major N-glycans identified in the tick samples, carcass and salivary glands, in different feeding stages. The names of the samples indicate male/female (F or M), salivary glands/carcasses (SG or CC), and before the onset of feeding, two days after, and five days after feeding attachments (0dAF, 2dAF, and 5dAF). The number in each cell is for relative percent abundance indicated by the blue bar. ID of each glycan is shown by the molecular weight (*m*/*z*). Color codes for the ID are as follows: the ID with blue background is for the glycan containing galactose-alpha-1,3-galactose (aGal), red background is for the glycan containing xylosylation, and black background in is for the glycan containing both aGal and xylosylation. In this analyses, same molecular weights containing mix of two different glycan were corrected by normalization factor ½ in the percent quantity. See the data with the mix of more than one glycan for same ID (molecular weight) which are found in the Appendix A. Structures of the glycan were shown for the glycans with more than 5% of total ion. For details of entire data in a worksheet, please see Appendix A.

**Figure 2 vaccines-08-00018-f002:**
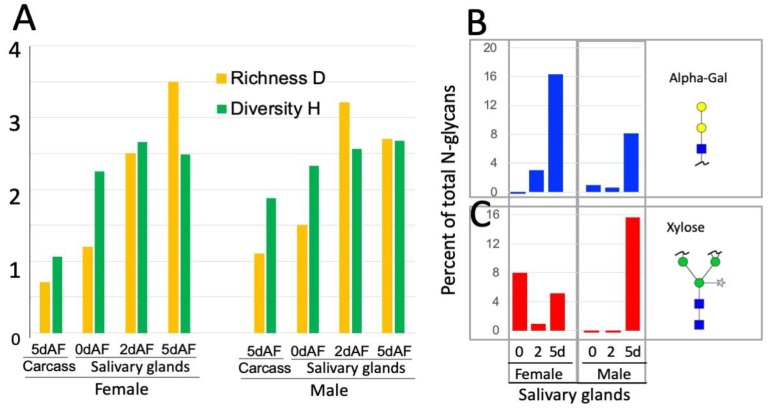
Dynamic changes in the N-glycan shown by (**A**) the richness D and diversity H, (**B**) changes in aGal glycans, and (**C**) the glycans with the core xylose during the time of feeding in the tick salivary glands.

**Figure 3 vaccines-08-00018-f003:**
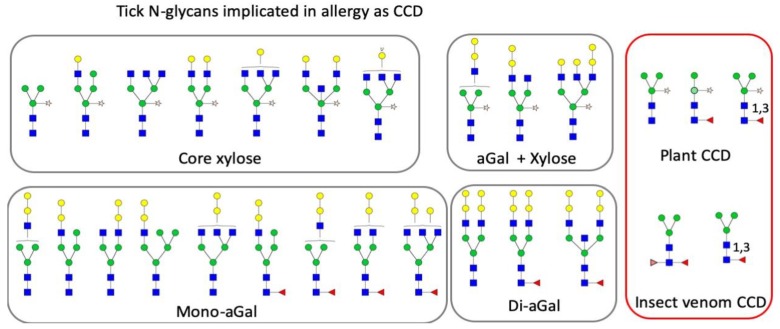
N-glycans identified in the salivary glands of *Amblyomma americanum*, containing the core xylose and terminal aGal, and both. Right column shows cross-reactive carbohydrate determinants (CCDs) described for plants and insect venom. Note that these tick N-glycan are implicated in allergy by containing the CCDs.

**Figure 4 vaccines-08-00018-f004:**
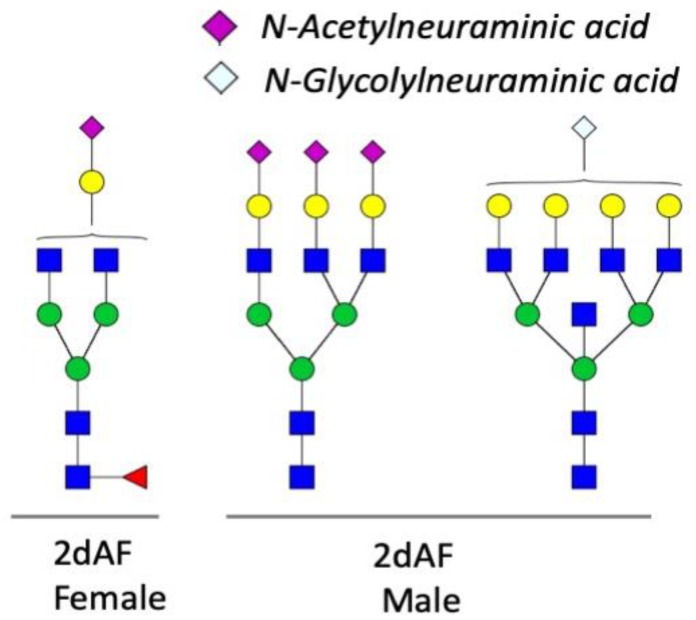
Trace N-glycans carrying the terminal *N-Acetylneuraminic acid* and *N-Glycolylneuraminic acid* in the salivary glands of *Amblyomma americanum*.

**Figure 5 vaccines-08-00018-f005:**
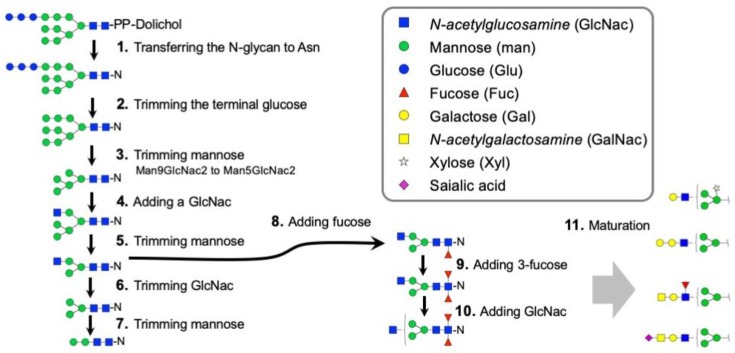
Generalized biochemical steps in biosynthesis of N-glycans. The numbers in each metabolic step involve the enzymes shown in Figure 6 and Figure 7 with the matching numbers.

**Figure 6 vaccines-08-00018-f006:**
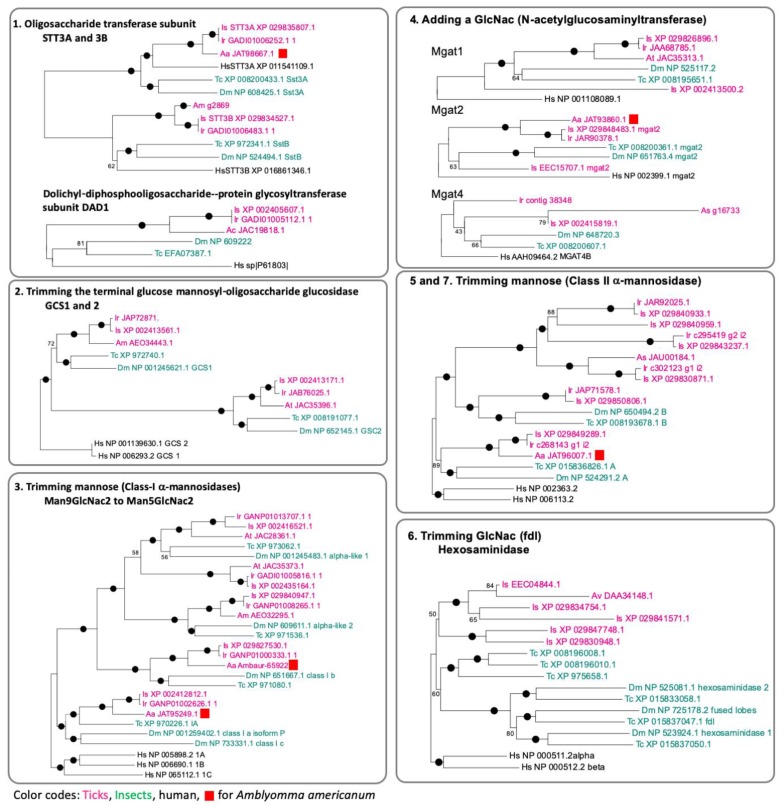
Phylogenetic trees for the genes involved in the N-glycan biosynthesis at the early Steps 1 to 7 in Figure 5. The numbering and the biological activities are referenced in Figure 5. Prefixes indicate the species: Aa for *Amblyomma americanum*, Ac for *A. cajennense*, Am for *A. meculatum*, As for *A. sculptum*, At for *A. triste*, Av for *A. variegatum*, Dm for *Drosophila melanogaster*, Hs for *Homo sapience*, Ir for *Ixodes ricinus*, Is for *I. scapularis*, and Tc for *Tribolium castaneum*. Color codes are for the genes with the fonts: magenta for ticks, green for insects, and black for human. Aa (*Amblyomma americanum)* is highlighted by red square. Note that Aa sequence in the database is yet incomplete and likely caused the low levels of coverage of the subjected genes in the analyses.

**Figure 7 vaccines-08-00018-f007:**
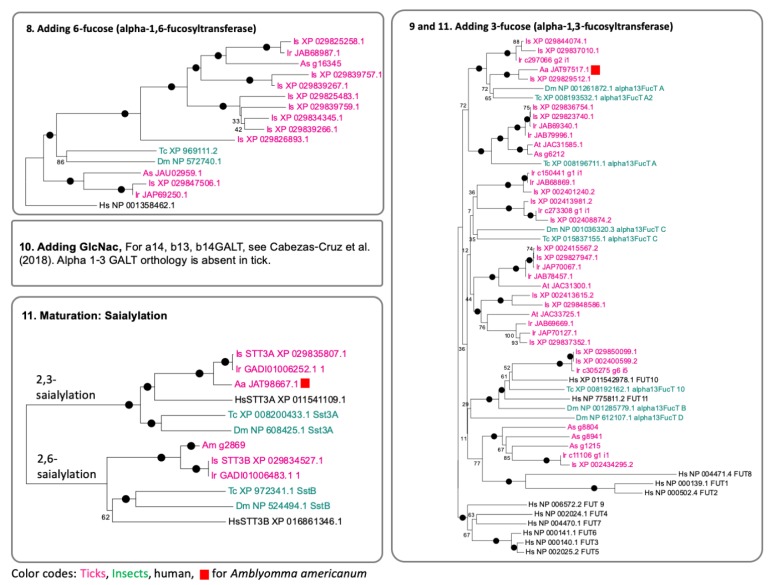
Phylogenetic trees for the genes involved in the N-glycan biosynthesis at the early Steps 9 to 11 in Figure 5. See figure caption for Figure 6 for the details.

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
