# Peer review of "Alpha-Gal and Cross-Reactive Carbohydrate Determinants in the N-Glycans of Salivary Glands in the Lone Star Tick, Amblyomma americanum"

_vaccines, 2020, doi:10.3390/vaccines8010018_

Round 1

Reviewer 1 Report

Park et al demonstrate a profile of N-glycans of the salivary glands in the feeding lone star tick, Amblyomma americanum. Lone star tick bites have been implicated in cases of human red meat allergy (RMA) in the SE United States, marked by an antibody response to galactose alpha-1,3-galactose (aGal) carbohydrates abundant in non-primate tissues. Sensitization of humans to aGal may be further enhanced by additional allergenic components of tick saliva proteins, including xylosylation of core-mannose or insect fucosylation of basal GlcNAc. With additional analysis of the genetics underlying tick N-glycosylation enzymes, the authors contribute to a robust framework for further research on a testable hypothesis of RMA.

The manuscript is divided into two phases, 1) N-glycan profiling of tick whole carcases and salivary glands at stages of feeding, and 2) phylogenetic analysis of tick N-glycosylation genes to determine the evolutionary diverged pathways that underlie tick salivary gland aGal generation. The manuscript logic is clear and the text is fairly well written, but has some issues of clarity that could be addressed to improve presentation.

# How do I reconcile the presentation of aGal abundance between presentations in Figures 1 and 2B? Figure 1 tells me that several aGal glycans comprise upwards of 15% of ions in M/F SG 2dAF (2489.3), yet these represent less than 3% of total ‘N-glycans’? I assume this is a difference of analysis in Figure 2B that only accounts for a subset of ions from the profile (ie change in the denominator)? Yet, on line 174-176 the authors state this glycan contributed to increasing the diversity and proportion of N-glycans… then I would expect that such an analysis would enrich aGal % in the dataset. I would encourage the authors to clarify the analysis done for Figure 2B for myself and perhaps be more explicit in the text. This extends to a lack of clarity as to H and D analysis parameters used.

# I primarily feel the figures need to be improved for clarity and ease of presentation for the reader. Particularly, captions should best match the image.
Figure 1, caption: Carcass, salivary glands, and the term ‘days after feeding’ should all denote their abbreviation as seen in the figure image. Define color code of aGal, xylo-, and both.
Figure 2: There is no C.
Figure 4: I would argue this is an unnecessary figure, the reporting of silaylated glycans in the text being sufficient without an image cartoon of the forms.
Figure 6/7: two suggestions: color coded legends for tick/insect/human directly in images, further denotation of Aa to draw the eye, perhaps with a symbol or boxing the name/code.
SFigs2-5: include abbreviations of samples (F_SG_2dAF) for continuity, in some cases the arrows to denote glycan cartoons bleed or obscure the peaks themselves. Decrease weight and/or use dashed lines.

# Samples: were all ticks utilized in pools (50 paired SG, 20 CC) from the same lot and experimental feeding protocol event? Was this one large experimental cohort with single sampling endpoints, or pools of some varied experimental cohorts?

# Per the conclusion that tick SG N-glycans may serve as molecular mimics in typical non-primate mammalian hosts… are there known/reported glycan profiles of major lone star tick hosts (deer, etc) which are endemic to the regions where RMA have been observed in the SE United States?

# Line 43-48: The introduction, as written, of the evolutionary underpinning of glycan diversity in varied species (being due to selective pressures) is under-supported in the text without clear references.

# Line 253: The statement of ‘rapid evolution’ in conjunction with tick expansion of several glycan enzymes seems presumptive and unsupported with the limited bioinformatic dataset available. I believe it is clear that there have been gene expansions in the tick lineages with divergence. I suggest a revised statement.

# Line 303-305: I am confused as to the nature and relevance of false positive sIgE detection of CCDs as stated in this current sentence construction.

Minor issues:

Line 88: the authors refer to humans as ‘non-natural hosts.’ I would suggest ‘atypical host.’ (tick bites of humans do occur in nature, as it were)

There are further minor copyediting issues that should be addressed, including:
Line 67, ‘sialic’; line 68, plural of nouns; line 303, ‘I’; throughout, aGal vs aGAL.

Author Response

Reviewer #1

Park et al demonstrate a profile of N-glycans of the salivary glands in the feeding lone star tick, Amblyomma americanum. Lone star tick bites have been implicated in cases of human red meat allergy (RMA) in the SE United States, marked by an antibody response to galactose alpha-1,3-galactose (aGal) carbohydrates abundant in non-primate tissues. Sensitization of humans to aGal may be further enhanced by additional allergenic components of tick saliva proteins, including xylosylation of core-mannose or insect fucosylation of basal GlcNAc. With additional analysis of the genetics underlying tick N-glycosylation enzymes, the authors contribute to a robust framework for further research on a testable hypothesis of RMA.

The manuscript is divided into two phases, 1) N-glycan profiling of tick whole carcases and salivary glands at stages of feeding, and 2) phylogenetic analysis of tick N-glycosylation genes to determine the evolutionary diverged pathways that underlie tick salivary gland aGal generation. The manuscript logic is clear and the text is fairly well written, but has some issues of clarity that could be addressed to improve presentation.

# How do I reconcile the presentation of aGal abundance between presentations in Figures 1 and 2B? Figure 1 tells me that several aGal glycans comprise upwards of 15% of ions in M/F SG 2dAF (2489.3), yet these represent less than 3% of total ‘N-glycans’? I assume this is a difference of analysis in Figure 2B that only accounts for a subset of ions from the profile (ie change in the denominator)? Yet, on line 174-176 the authors state this glycan contributed to increasing the diversity and proportion of N-glycans… then I would expect that such an analysis would enrich aGal % in the dataset. I would encourage the authors to clarify the analysis done for Figure 2B for myself and perhaps be more explicit in the text. This extends to a lack of clarity as to H and D analysis parameters used.

Response: We found that the misunderstanding was caused by missing the description of the procedure normalization for the m/z having mixed two glycans. The second paragraph of the Results now explains it as,

 “In this analyses, same molecular weights containing mix of two different glycan were corrected by normalization factor ½ in the percent quantity (i.e., 2244.1 and 2489.3)”.

# I primarily feel the figures need to be improved for clarity and ease of presentation for the reader. Particularly, captions should best match the image.
Figure 1, caption: Carcass, salivary glands, and the term ‘days after feeding’ should all denote their abbreviation as seen in the figure image. Define color code of aGal, xylo-, and both.

Response: Figure 1 caption is now changed to explain abbreviations and the color codes. Below, changed caption, shows the gray background for the added sentences.

“Fig. 1. Major N-glycans identified in the tick samples, carcass and salivary glands, in different feeding stages. The names of the samples indicate male/female (F or M), salivary glands/carcasses (SG or CC), and before the onset of feeding, 2 days after, and 5 days after feeding attachments (0dAF, 2dAF, and 5dAF). The number in each cell is for relative percent abundance indicated by the blue bar. ID of each glycan is shown by the molecular weight (m/z). Color codes for the ID are: the ID with blue background is for the glycan containing aGal, red background is for the glycan containing xylosylation, and black background in is for the glycan containing both aGal and xylosylation. In this analyses, same molecular weights containing mix of two different glycan were corrected by normalization factor ½ in the percent quantity. See the data with the mix of more than one glycan for same ID (MW) which are found in the Supporting information 1. Structures of the glycan were shown for the glycans with more than 5% of total ion.  For details of entire data in a worksheet, please see Supporting Fig. 1.”

Figure 2: There is no C.

Response: Figure 2: The C is now added.

Figure 4: I would argue this is an unnecessary figure, the reporting of silaylated glycans in the text being sufficient without an image cartoon of the forms.

Response: We partially agree with reviewer’s point for the unnecessary figure.  However, when we consult with other general readers who are not familiar with glycobiology, we received a feedback for the importance of presenting the structure showing the site of sialylation. Therefore, we leave the Figure as it is now unless the reviewer has a strong opinion for removal of this figure.

Figure 6/7: two suggestions: color coded legends for tick/insect/human directly in images, further denotation of Aa to draw the eye, perhaps with a symbol or boxing the name/code.
SFigs2-5: include abbreviations of samples (F_SG_2dAF) for continuity, in some cases the arrows to denote glycan cartoons bleed or obscure the peaks themselves. Decrease weight and/or use dashed lines.

Response: Both suggestions are incorporated into the current figures 6 and 7. Color codes are embedded on the image and the Aa (Amblyomma americanum) is highlighted by red squares. The figure caption is now modified by adding,

“Aa is highlighted by red square. Note that Aa sequence in the database is yet incomplete and likely caused the low levels of coverage of the subjected genes.”

SFigs2-5: include abbreviations of samples (F_SG_2dAF) for continuity, in some cases the arrows to denote glycan cartoons bleed or obscure the peaks themselves. Decrease weight and/or use dashed lines.

Response: Abbreviations in the figure captions are modified for including the abbreviations. The arrows indicating small peaks are drawn again for the clarity (dashed, red color, and decreased weight).

# Samples: were all ticks utilized in pools (50 paired SG, 20 CC) from the same lot and experimental feeding protocol event? Was this one large experimental cohort with single sampling endpoints, or pools of some varied experimental cohorts?

Response: Following sentence is added in the first paragraph of the Materials and Methods to clarify the sampling procedure.

“Dissected tissues from ~50 individual ticks were made over a duration of a month in multiple feeding chambers, and were stored at -80°C until the analyses.”

# Per the conclusion that tick SG N-glycans may serve as molecular mimics in typical non-primate mammalian hosts… are there known/reported glycan profiles of major lone star tick hosts (deer, etc) which are endemic to the regions where RMA have been observed in the SE United States?

Response: The sentence was rephrased and two references was added as,

“…in the case of parasitization of non-primate mammals (i.e., goat, sheep, horse, mouse, pig, and deer) that use aGal in the intrinsic system [42,43].

Ogawa, H.; Galili, U. Profiling terminal N-acetyllactosamines of glycans on mammalian cells by an immuno-enzymatic assay. Glycoconjugate J 2006, 23, 663-674, doi:10.1007/s10719-006-9005-0. Tan, Y.X.; Gong, F.; Li, S.B.; Ji, S.P.; Lu, Y.P.; Gao, H.W.; Xu, H.; Zhang, Y.P. Brief report: a new profile of terminal N-acetyllactosamines glycans on pig red blood cells and different expression of alpha-galactose on Sika deer red blood cells and nucleated cells. Glycoconjugate J 2010, 27, 427-433, doi:10.1007/s10719-010-9289-y.”

# Line 43-48: The introduction, as written, of the evolutionary underpinning of glycan diversity in varied species (being due to selective pressures) is under-supported in the text without clear references.

Response: The sentence was rephrased and a reference was added as,

“…, whereas the parts of N-glycan variations were consequences of selection pressure in the arms races between pathogen and immune systems [7].”

Reference 7, Gagneux, P.; Aebi, M.; Varki, A. Evolution of Glycan Diversity. In Essentials of Glycobiology, 3rd ed.; Varki A, C.R., Esko JD, et al., Ed. Cold Spring Harbor Laboratory Press: Cold Spring Harbor (NY), 2017; doi: 10.1101/glycobiology.3e.020”

# Line 253: The statement of ‘rapid evolution’ in conjunction with tick expansion of several glycan enzymes seems presumptive and unsupported with the limited bioinformatic dataset available. I believe it is clear that there have been gene expansions in the tick lineages with divergence. I suggest a revised statement.

Response: A sentence was added to clarify the point as,

“This conclusion is yet based on the gene repertoire of I. scapularis which provided the most comprehensive gene sets in the genome project, while other tick species also show general patterns supporting the conclusion, but with limited numbers of the search results”.

# Line 303-305: I am confused as to the nature and relevance of false positive sIgE detection of CCDs as stated in this current sentence construction.

Response: The sentence was rephrased as,

“However, clinical importance of the CCD as the causal factors of allergy is yet unclear, while correlation between sIgE activities against the specific CCD and the clinical symptoms has been shown [30,48].”

Minor issues:

Line 88: the authors refer to humans as ‘non-natural hosts.’ I would suggest ‘atypical host.’ (tick bites of humans do occur in nature, as it were)

It is now changed to atypical host in two places.

There are further minor copyediting issues that should be addressed, including:
Line 67, ‘sialic’; line 68, plural of nouns; line 303, ‘I’; throughout, aGal vs aGAL.

Four changes mentioned above are corrected as the reviewer suggested.

Reviewer 2 Report

Authors Yoonseong Park et al, made an attempt to study the role of alpha-gal and cross-reactive carbohydrate in the N-glycans of salivary glands in the lone star tick, Amblyomma americanum

This work is very well planned and the experiments were performed well. The results are presented in 7 figures and 5 supplemental figures. Over all the results are discussed appropriately and concluded.

Please address the following minor comments

Materials and methods : Please provide references for methods for line # 109 and 119 Line # 261:  (reviewed in [2,33]; please close the parentheses  Line # 303: "However, it I also debated that" please correct this. Font size difference in different paragraphs e.g. Line #34 to Line #49.

Author Response

Reviewer #2

Authors Yoonseong Park et al, made an attempt to study the role of alpha-gal and cross-reactive carbohydrate in the N-glycans of salivary glands in the lone star tick, Amblyomma americanum

This work is very well planned and the experiments were performed well. The results are presented in 7 figures and 5 supplemental figures. Over all the results are discussed appropriately and concluded.

Please address the following minor comments

Materials and methods : Please provide references for methods for line # 109 and 119

A reference (reference 25) for the method is added as,

“Shajahan, A.; Heiss, C.; Ishihara, M.; Azadi, P. Glycomic and glycoproteomic analysis of glycoproteins-a tutorial. Anal Bioanal Chem 2017, 409, 4483-4505, doi:10.1007/s00216-017-0406-7”

Line # 261:  (reviewed in [2,33]; please close the parentheses 

Change is made in the revision.

Line # 303: "However, it I alsodebated that" please correct this.

This sentence is now rephrase to soften the meaning (see also the reviewer #1) as,

“However, clinical importance of the CCD as the causal factors of allergy is yet unclear, while correlation between sIgE activities against the specific CCD and the clinical symptoms has been shown [30,48].”

Font size difference in different paragraphs e.g. Line #34 to Line #49. 

The sizes of the fonts are changed to make them same.